Citation: *Molecular Systems Biology* 9:667
www.molecularsystemsbiology.com

# The *Neurospora* photoreceptor VIVID exerts negative and positive control on light sensing to achieve adaptation

Elan Gin[1,2], Axel CR Diernfellner[3], Michael Brunner[3,*] and Thomas Höfer[1,2,*]

[1] Division of Theoretical Systems Biology, German Cancer Research Center (DKFZ), Heidelberg, Germany, [2] Bioquant Center, University of Heidelberg, Germany and [3] University of Heidelberg Biochemistry Center (BZH), Heidelberg, Germany
* Corresponding authors. M Brunner, University of Heidelberg Biochemistry Center (BZH), Im Neuenheimer Feld 328, Heidelberg 69120, Germany.
Tel.: +49 6221 544207; Fax: +49 6221 544769; E-mail: michael.brunner@bzh.uni-heidelberg.de or T Höfer, Division of Theoretical Systems Biology, German Cancer Research Center (DKFZ), Im Neuenheimer Feld 280, Heidelberg 69120, Germany. Tel.: +49 6221 5451380; Fax: +49 6221 5451487; E-mail: t.hoefer@dkfz-heidelberg.de

The light response in *Neurospora* is mediated by the photoreceptor and circadian transcription factor White Collar Complex (WCC). The expression rate of the WCC target genes adapts in daylight and remains refractory to moonlight, despite the extraordinary light sensitivity of the WCC. To explain this photoadaptation, feedback inhibition by the WCC interaction partner VIVID (VVD) has been invoked. Here we show through data-driven mathematical modeling that VVD allows *Neurospora* to detect relative changes in light intensity. To achieve this behavior, VVD acts as an inhibitor of WCC-driven gene expression and, at the same time, as a positive regulator that maintains the responsiveness of the photosystem. Our data indicate that this paradoxical function is realized by a futile cycle that involves the light-induced sequestration of active WCC by VVD and the replenishment of the activatable WCC pool through the decay of the photoactivated state. Our quantitative study uncovers a novel network motif for achieving sensory adaptation and defines a core input module of the circadian clock in *Neurospora*.
*Molecular Systems Biology* **9**: 667; published online 28 May 2013; doi:10.1038/msb.2013.24
*Subject Categories:* metabolic and regulatory networks; signal transduction
*Keywords:* adaptation; mathematical model; *Neurospora*; protein–protein interaction; VVD

## Introduction

Circadian clocks are a fundamental adaptation to the day–night rhythm in all kingdoms of life. They are entrained by light cues from the environment, whose intensity varies due to the chosen habitat and fluctuates randomly because of weather conditions. Experimental work on the model organism *Neurospora crassa* indicates that its clock does not perceive directly the fluctuating environmental light intensity but, rather, that the light is processed by an input module of the clock that exhibits sensory adaptation (Schwerdtfeger and Linden, 2001; Heintzen *et al*, 2001; Malzahn *et al*, 2010). The idea of a light-processing unit has previously been suggested on theoretical grounds (Kronauer *et al*, 1999), but its molecular basis is unknown.

The circadian clock of the fungus *Neurospora crassa* exemplifies the transcriptional negative feedback design found in a multitude of organisms (Brunner and Káldi, 2008; Baker *et al*, 2012). Here the clock proteins White Collar 1 and 2 (WC-1 and WC-2) act as global activators of photoresponses. WC-1, a blue-light photoreceptor, and WC-2 form the transcription factor White Collar Complex (WCC) that drives the expression of target genes with ∼24 h periodicity. Among many other genes, WCC activates the expression of the clock gene *frequency* (*frq*); FRQ protein in turn inhibits the WCC. This negative feedback loop is thought to generate the circadian cycles of WCC activity in constant darkness—the hallmark of an autonomous clock (Gonze *et al*, 2000; Tseng *et al*, 2012).

The *Neurospora* light response exhibits sensory adaptation, rendering WCC-driven gene expression transient in constant light (Schwerdtfeger and Linden, 2001; Heintzen *et al*, 2001). Recently, Malzahn *et al* (2010) demonstrated directly that photoadaptation prevents the resetting of the circadian clock in *Neurospora* by moonlight. Strikingly, a mutant strain with defective photoadaptation has such high light sensitivity that it perceives moonlight and fails to entrain to natural day–night cycles. This finding shows that besides short-term adaptation, the *Neurospora* light response displays longer-term memory of experienced daylight that makes the organism insensitive to light input during the night and at dawn (Elvin *et al*, 2005; Malzahn *et al*, 2010).

Two kinds of mechanisms have been invoked to explain *Neurospora* photoadaptation: constitutive removal of light-activated WCC (Talora *et al*, 1999; He and Liu, 2005; Tsumoto *et al*, 2011) and feedback inhibition through the blue-light

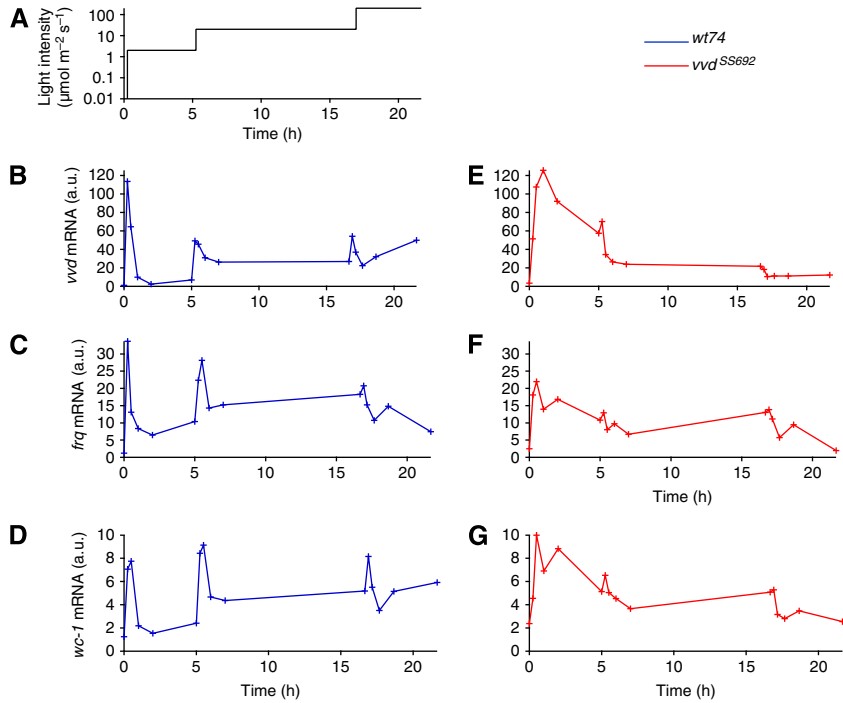

**Figure 1** Rapid adaptation and repeated induction depend on VVD. Light induction experiments for *vvd*, *wc-1* and *frq*. Liquid cultures of the *vvd* loss-of-function mutant (*vvd*[SS69]) and its corresponding wild-type strain (*wt74*) were raised in light for 24 h and then transferred to constant darkness for 24 h before light induction. Initial light intensity was 2 μmol m$^{-2}$ s$^{-1}$ and was raised to 20 μmol m$^{-2}$ s$^{-1}$ after 300 min and again to 200 μmol m$^{-2}$ s$^{-1}$ after 1000 min (**A**). Samples were collected at the indicated time points and mRNA levels for *vvd*, *frq* and *wc-1* were measured via qRT–PCR. The wild-type *vvd* mRNA levels at $t = 0$ were set to 1. To compare the relative levels of the mRNAs, we corrected for the efficiency of the various RT–PCR probes used, by performing qPCR with a dilution series of genomic DNA. The values were corrected with reference to *vvd* (further details in the Supplementary Information). Wild-type data are shown in panels (**B–D**) and vvdSS692 mutant data are shown in panels (**E–G**). Throughout the paper, wild-type data are shown in blue and *vvd*[SS692] mutant data are shown in red. Source data for this figure is available on the online supplementary information page.

photoreceptor VIVID (VVD) (Heintzen *et al*, 2001; Schwerdtfeger and Linden, 2001, 2003; Shrode *et al*, 2001). By now, a large body of work has shown that photoadaptation in *Neurospora* requires the blue-light photoreceptor VVD (reviewed in Baker *et al*, 2012). VVD is induced by light-activated WCC and inhibits the function of the latter as a transcriptional activator. The combination of rapid induction and slow degradation of VVD might realize the adaptation of the photoresponse during the day and prolonged refractoriness during the night.

VVD physically interacts with WC-1 and represses the transcription of WCC target genes (Chen *et al*, 2010; Hunt *et al*, 2010; Malzahn *et al*, 2010). Specifically, the light-activated WCC complex dimerizes on the light-response elements (LREs), such as those found in the *frq* promoter, via an interaction of the photosensory light–oxygen–voltage (LOV) domains of WC-1. Light-activated VVD, which also harbors a LOV domain, disrupts WCC homodimerization by competitive binding to WC-1 (Malzahn *et al*, 2010). Thus, VVD acts as a feedback inhibitor of light-induced transcription by sequestering the active WCC. However, this downregulation of the light response is only one aspect of photoadaptation. At the same time, *Neurospora* can respond to further increases in light intensity, perceiving intensity changes rather than the absolute light level (Schwerdtfeger and Linden, 2003; Chen *et al*, 2010; Malzahn *et al*, 2010). As fungi may inhabit sites with very different ambient light intensities (e.g., underneath tree bark versus sun-exposed surfaces), the sensing of relative light changes will be crucial for the proper functioning of the circadian clock.

From a mechanistic point of view, however, it is perplexing that feedback inhibition of the WCC by VVD should bring about this finely tuned adaption behavior. If VVD efficiently sequesters the central activator of the photoresponse, one must ask how *Neurospora* stays responsive to increases in light intensity. This is a general problem, because adaptation involves tonic inhibition that tends to downregulate the sensitivity of a sensory system. Moreover, mathematical models have shown that direct deactivation of a positive regulator generally fails to produce good adaptation (Behar *et al*, 2007; Ma *et al*, 2009), further questioning a simple inhibition model.

To dissect the molecular mechanisms of photoadaptation in *Neurospora* on a quantitative basis, we have combined mathematical modeling with systematic, time-resolved measurements of the expression of the core genes, *wc-1*, *frq* and *vvd*. We identified a novel 'adaptation motif' in the WCC–FRQ–VVD reaction network and suggest that this motif allows *Neurospora* to sense relative changes in light intensity. This motif employs the futile cycling of photoactivated states to turn simple feedback inhibition into adaptation. Thus, our study defines an essential, non-oscillatory module of the circadian clock in *Neurospora*. In addition, it identifies a hitherto unrecognized network motif for achieving sensory adaptation.

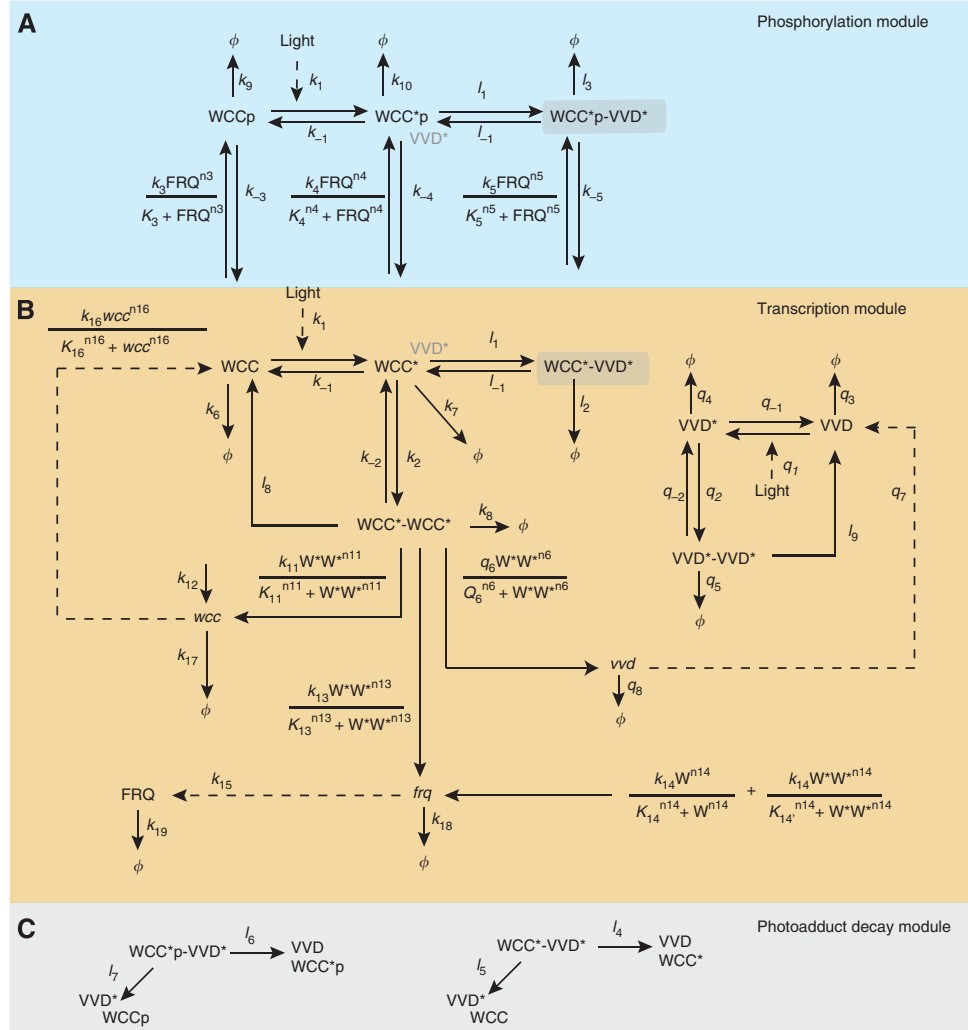

**Figure 2** Model scheme. Protein variables are indicated in normal font and mRNA variables are italicized. Light-activated forms of protein are denoted with an asterix. Phosphorylated states are of the form WCCp. The model is described in three modules: a phosphorylation module (**A**, shaded blue) involving FRQ-dependent phosphorylation of the WCC; a transcription module (**B**) that includes the light activation of WCC, subsequent homodimerization and transcription; and the photoadduct decay module (**C**, shaded gray). The photoadduct decay module shows the additional pathways for breakdown of the heterodimer complex.

# Results

## Rapid photoadaptation and continued light responsiveness of *Neurospora* depend on VVD

To quantify the light-sensing ability of *Neurospora* and systematically probe the role of VVD, we subjected the wild-type strain (*wt74*) and the corresponding VVD loss-of-function mutant *vvd⁻* (*vvd^SS692*, Heintzen *et al*, 2001) to three increasing light steps on a logarithmic scale, from very dim light to the light intensity of a cloudy day (2, 20 and 200 μmol photons $m^{-2} s^{-1}$, Figure 1A). The prolonged exposure to constant light intensity was chosen to quantify the transient light response and subsequent return to a steady state. The *vvd^SS692* mutant allele is transcribed but no functional protein is produced, allowing us to compare the transcription of the *vvd* gene in the presence (wild type) and absence (mutant) of VVD protein. Normalizing to *vvd* ($t = 0$) in the wild type and correcting for the efficiency

of the various RT–PCR probes (Supplementary Figure 1), we found that *vvd* was generally the most abundant mRNA species after light induction followed by *frq* and *wc-1* (Figure 1A–G).

After the transfer from dark to the lowest light intensity ($t = 0$), the levels of all three mRNAs peaked within 30 min and then decreased rapidly in the wild-type strain (Figure 1B–D), and much more slowly in the mutant (Figure 1E–G). Subsequent light steps evoked further responses in the wild type, showing that the photosystem remains responsive to higher light intensities (Figure 1B–D; *cf*. Schwerdtfeger and Linden, 2003; Malzahn *et al*, 2010). By contrast, responsiveness to subsequent light steps was strongly diminished in the *vvd⁻* mutant (Figure 1E–G). Taken together, these data show that VVD has both an inhibitory function in light sensing, causing rapid inactivation of the photoresponse in constant light, and a positive role in mediating continued responsiveness to higher light intensities.

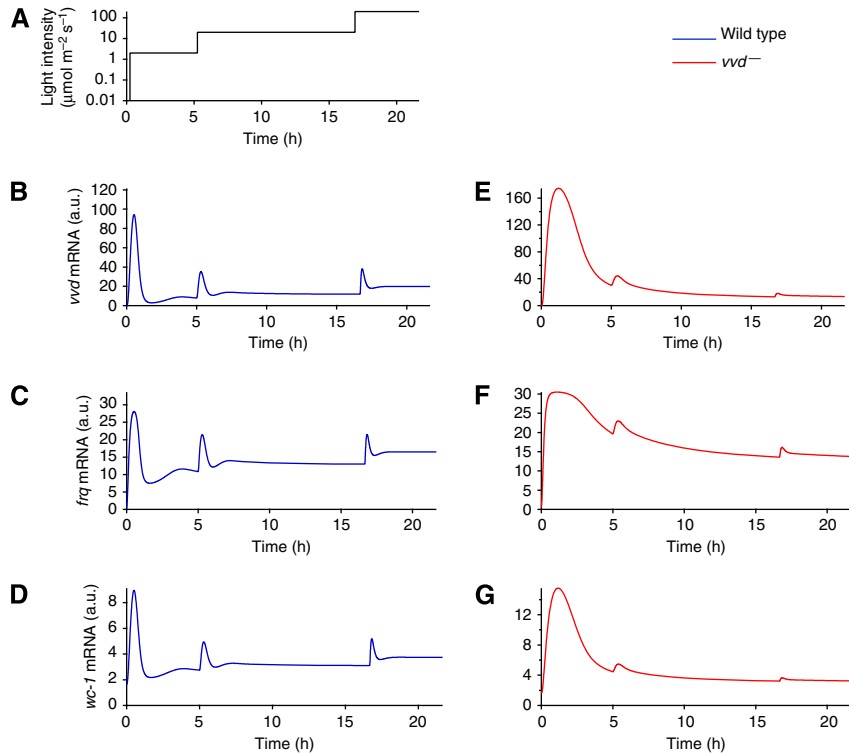

**Figure 3** The model accounts for the photoresponse dynamics of the wild type and mutant. The model scheme was fitted to the light induction experiments with the light regime shown in (**A**), and, consistent with the experimental data, the wild type shows repeated activation in response to increasing light (**B–D**), while responsiveness is diminished in the mutant (**E–G**).

## Mathematical model for *Neurospora* light sensing and adaptation

To understand how VVD can act as both an inhibitor and a positive regulator, we integrated the key molecular interactions described experimentally into a mathematical model (Figure 2). The core part of this model is the transcription module (Figure 2B) that describes the expression of the WCC target genes, *vvd*, *frq* and *wc-1*, and the regulation of the transcriptional activity of WCC by light and by VVD. Light converts the dark forms of the WCC and VVD into the light-activated forms WCC* and VVD* through the formation of a photoadduct in their respective LOV domains. The resulting conformational change of the proteins allows the homodimerization and heterodimerization of WCC* and VVD* (Zoltowski and Crane, 2008; Zoltowski *et al*, 2009; Malzahn *et al*, 2010; Vaidya *et al*, 2011; Schafmeier and Diernfellner, 2011). Light-activated WCC* homodimers activate the transcription of *wc-1*, *vvd* and *frq* by binding to LREs. In the dark, transcription of *wc-1* is driven by a basal rate independent of the WCC, while the dark-form WCC drives the transcription of *frq* by binding to the Clock box (C-box) of its promoter (Froehlich *et al*, 2002, 2003), but not transcription of *vvd* (Káldi *et al*, 2006; Hunt *et al*, 2007). The WCC*–VVD* heterodimer is transcriptionally inactive (Malzahn *et al*, 2010). Therefore, VVD functions as a feedback inhibitor of the light-activated WCC. The expression of WCC, VVD and FRQ is balanced by protein and mRNA degradation. The light-activated, transcriptionally active

WCC* homodimer is degraded rapidly, whereas the various VVD-bound and dark-form WCC species are more stable (He *et al*, 2005; Schafmeier *et al*, 2006; Schafmeier *et al*, 2008; Malzahn *et al*, 2010).

The phosphorylation module (Figure 2A) describes an additional negative feedback loop on WCC activity through FRQ-mediated phosphorylation that contributes to controlling WCC-driven gene expression. In constant dark, it can generate self-sustained circadian oscillations, involving complex regulation of WCC and FRQ by phosphorylation, nucleocytoplasmic shuttling and degradation (Merrow *et al*, 2001; He *et al*, 2005; Schafmeier *et al*, 2005; Cha *et al*, 2008; Schafmeier *et al*, 2008; Diernfellner *et al*, 2009; Querfurth *et al*, 2011; Diernfellner and Schafmeier, 2011, Tseng *et al*, 2012).

The light-induced photoadducts in the LOV domains of WCC* and VVD* are metastable and revert to the dark form spontaneously (Guo *et al*, 2005). Therefore, the WCC*–VVD* heterodimer can decompose through distinct pathways: (i) dissociation of the light-activated heterodimers and (ii) photoadduct decay in one binding partner, upon which the partners lose affinity for one another. This latter mechanism, depicted in the photoadduct decay module (Figure 2C), allows for the 'futile cycling' of WCC*–VVD* heterodimer formation through photoactivation followed by photoadduct decay. For consistency, we also include the dissociation of the WCC* and VVD* homodimers triggered by photoadduct decay into the model.

We translated the reaction scheme into coupled ordinary differential equations for the concentrations of the 14 mRNA and protein species (Supplementary Text S1).

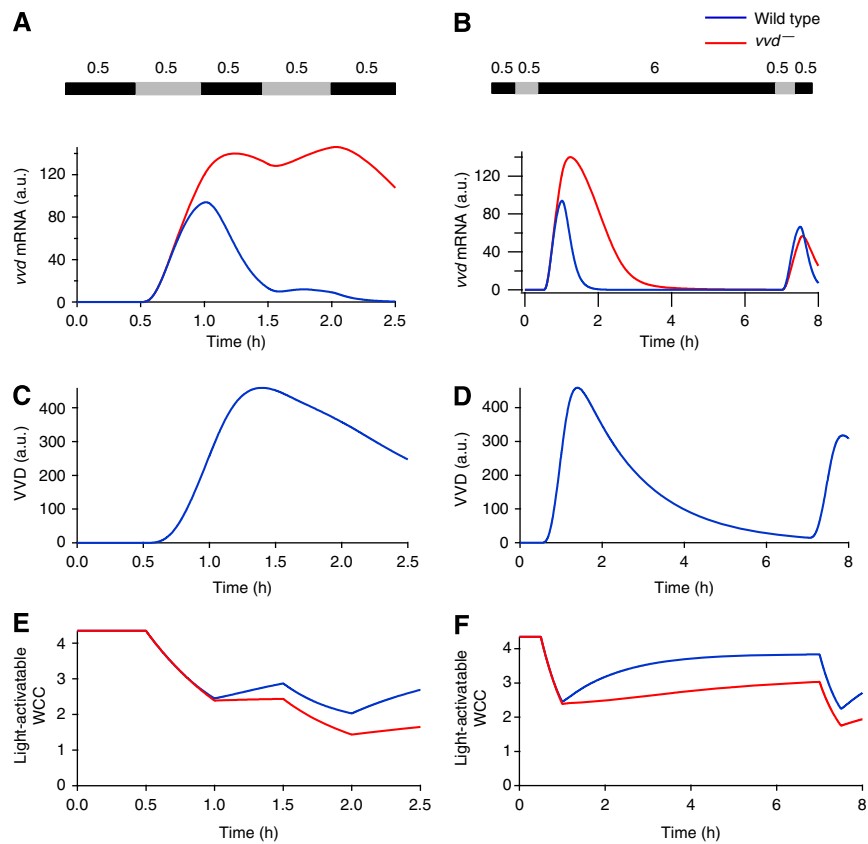

**Figure 4** VVD sets the refractory period after light response. Model-simulated responses to different dark/light regimes with light levels of 20 $\mu$mol m$^{-2}$ s$^{-1}$. The intervening dark incubation period between the two light pulses are 30 min (left panel) or 6 h (right panel). (**A** and **B**) After only 30 min of intervening darkness, the *vvd*⁻ mutant responds to the second light pulse, whereas the wild type requires a period of 6 h in the dark before it can respond again. (**C** and **D**) A substantial pool of VVD still remains after 30 min in the dark, thus preventing a response to the second light pulse in the wild type. After 6 h, the VVD pool is depleted, restoring full responsiveness. (**E** and **F**) Light-activatable WCC (dark form) levels recover faster in the wild type than the *vvd*⁻ mutant during both durations of dark period, illustrating the role of VVD in restoring the WCC. On exposure to the second light pulse, the WCC is activated in both the wild type and *vvd*⁻ mutant. However, the VVD remaining after 30 min of dark suppresses the light response, and only after 6 h of darkness are VVD levels low such that the wild type is able to respond to the light pulse.

## The model accounts for photoresponse dynamics in wild type and *vvd*⁻ mutant

To fit the model to the light-step experiments in Figure 1, we first determined the kinetic parameters for the model without VVD protein, mimicking the *vvd*⁻ mutant (using both a Bayesian Markov chain Monte Carlo algorithm and a maximum likelihood estimate; Supplementary Text S1). The available data constrain parameters of the model that govern VVD function (including VVD production, heterodimerization with WCC and photoadduct decay; Supplementary Figure S5). For a reduced, biologically less-detailed version of the model, all parameters can be identified from the data (Supplementary Text S2). The parameterized model reproduces the very slow decline of the response after the first light step, caused by degradation of light-activated WCC and inhibition of WCC by FRQ-mediated phosphorylation. The following two light steps do not elicit substantial responses (Figure 3E–G; *cf.* Figure 1E–G). Interestingly, the *vvd*⁻ mutant data show an immediate drop of mRNA levels after the light pulse that cannot be explained by the present model and hints at additional mechanisms regulating mRNA synthesis or degradation.

Next we introduced functional VVD protein to simulate the wild type. To fit the full model, we kept the kinetic parameters

of all processes not involving VVD unchanged from the *vvd*⁻ mutant model and only determined the VVD-related parameters. The introduction of VVD fully restored rapid photoadaptation and allowed repeated responses to increasing light steps (Figure 3A–D; *cf.* Figure 1A–D). Consistent with this positive role of VVD, the WCC protein levels were higher in the wild type than in the mutant (Supplementary Figure S2). Moreover, the model correctly accounts for *vvd* being the most abundant mRNA, followed by *frq* and *wc-1*.

In summary, the mathematical model reproduces salient features of *Neurospora* photoadaptation. Consistent with the experimental data, VVD mediates both rapid downregulation of the photoresponse in constant light and continued responsiveness to increases in light intensity.

## VVD sets refractory phase for light sensing

Next we asked whether the model parameterized with the light-step protocol exhibits a prolonged refractory phase that accounts for the unresponsiveness of the wild type to moonlight (Malzahn *et al*, 2010). We simulated successive transient light stimuli separated by a variable dark phase, displaying *vvd* mRNA as a measure of the transcriptional

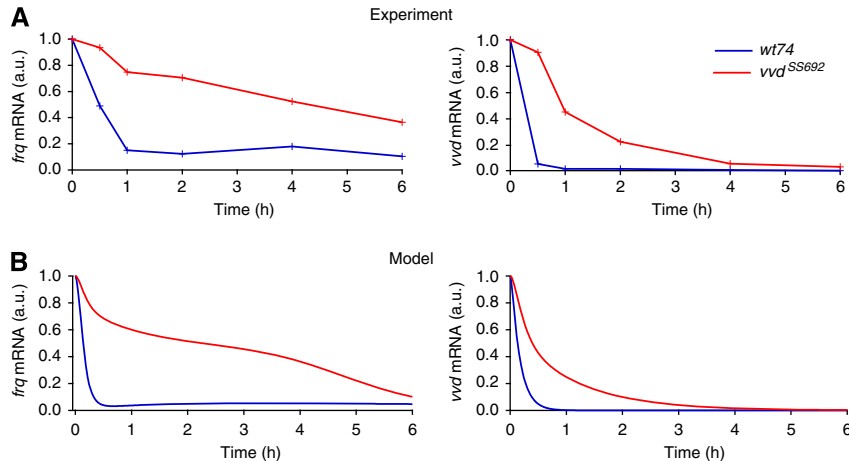

**Figure 5** VVD represses transcriptional activity after transfer to dark. (**A**) Repression of *frq* and *vvd* RNA synthesis upon light–dark transfer is slow in *vvd^SS692^*. Cultures of *wt* and *vvd^SS692^* were grown in constant light (20 µmol m$^{-2}$ s$^{-1}$) for 2 days, transferred to darkness and collected at the indicated time points. Levels of *vvd* and *frq* mRNA are shown as determined by real-time PCR. mRNA levels are normalized to the value at the light-to-dark transfer point ($t = 0$). (**B**) Simulations reproduce the experimental result showing *frq* and *vvd* mRNA levels elevated for up to 4 h in the *vvd⁻* mutant and wild-type levels reaching baseline within 1 h. Source data for this figure is available on the online supplementary information page.

response. In the wild type, VVD protein efficiently suppresses the response to a rapidly following second stimulus (Figure 4A and C). Responsiveness to the second stimulus is restored only after a dark incubation of ∼6 h (Figure 4B), when the VVD level has dropped (Figure 4D). The key role of VVD is highlighted by comparing the wild type with the *vvd⁻* mutant: the latter reacts to the second light stimulus both after short (30 min) and long (6 h) dark incubation (Figure 4A and B). These simulations match previous experiments (Arpaia *et al*, 1999, Schwerdtfeger and Linden, 2001), showing that the model correctly explains the prolonged suppression of WCC-mediated transcription by VVD.

Remarkably, the model predicts that the pool of WCC that can be activated by light (*cf.* Figure 2) recovers during the dark periods more quickly in the wild type than in the *vvd⁻* mutant (Figure 4E and F), further illustrating that VVD also has a positive effect on the WCC. Taken together, the model simulations show that the kinetics of VVD degradation in the dark set the refractory period during which the photosystem remains blind to low-light input. At the same time, VVD supports the recovery of the light-activatable WCC.

## VVD efficiently represses the transcriptional activity of light-activated WCC

Although VVD supports higher levels of WCC in the light, its inhibitory effect on the WCC should be dominant so as to suppress spurious responses to low-light input. Indeed, Hunt *et al* (2010) have reported that *frq* transcription in the wild-type strain was rapidly reduced to basal levels upon light–dark transfer, whereas *frq* mRNA remained elevated for several hours in a *vvd*-knockout strain. We corroborated this effect for *frq* mRNA. Furthermore, we observed it for *vvd* mRNA, albeit with somewhat different kinetics (Figure 5A). These data are consistent with the rapid inactivation of light-activated WCC by VVD. The model reproduces the data without parameter adjustment (Figure 5B). In particular, the lack of inhibition of

light-activated WCC by VVD explains prolonged transcription in the dark in the *vvd⁻* mutant.

The transcript level of *frq* declines more slowly than that of *vvd* in the mutant. Moreover, *frq* transcription in the wild type is maintained at a basal level in the long run, while *vvd* transcription vanishes (*cf.* Figure 5A). To explain this difference, we have implemented the different architectures of the *vvd* and *frq* promoters in the model. The *vvd* transcription is driven by light-activated WCC; *frq* transcription is activated by two promoter elements: the proximal LRE, mediating light induction, and the C-box, which can be activated by the dark-form WCC (Froehlich *et al*, 2002). This difference in gene regulation naturally accounts for the different kinetics of *vvd* and *frq* mRNA (*cf.* Figure 5B).

In prolonged darkness, self-sustained oscillations of the circadian clock are independent of VVD. The model reproduces this phenomenon (Supplementary Figure S3). In agreement with experiment (Elvin *et al*, 2005), the *vvd⁻* mutant shows a phase delay of ∼4 h.

To summarize, these results show that the competitive binding of VVD to WCC can fully account for the efficient quenching of the transcriptional activity of light-activated WCC. In this way, VVD produced during light exposure also times the onset of the free-running clock in constant darkness without being involved in generating the oscillations.

## Futile cycling turns feedback inhibition into adaptation

The quantitative agreement of the model with diverse experimental scenarios provides strong evidence for VVD's function as a competitive feedback inhibitor of the light-activated WCC. However, VVD is essential also for continuous light responsiveness of the photosystem (*cf.* Figure 1). Analyzing the dynamics of the WCC pool in the model, we observed that a futile cycle is the key to repeated light sensing (Figure 6A). This cycle is generated by the heterodimerization of light-activated WCC and VVD, followed by the spontaneous

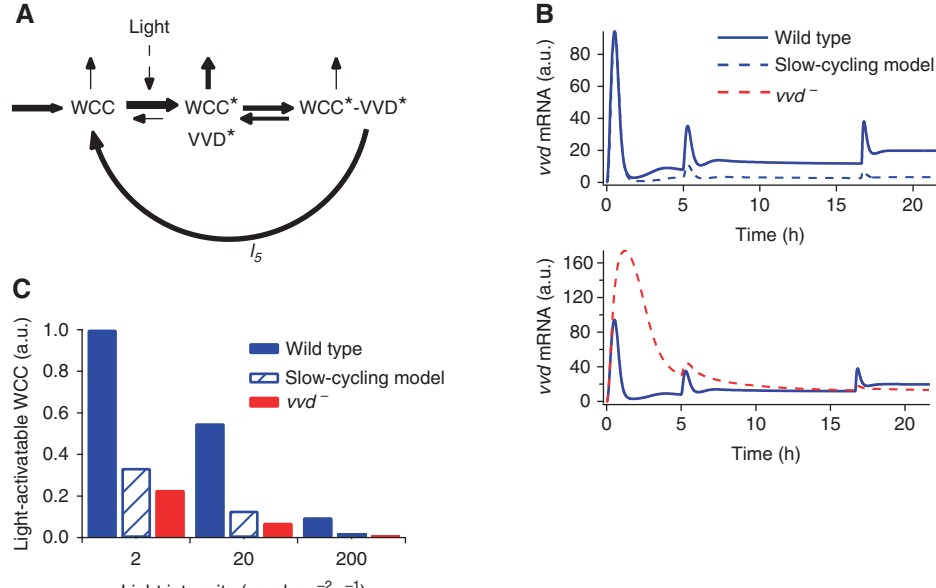

**Figure 6** Photoadduct decay constitutes a futile cycle that ensures repeated sensitivity. (**A**) Schematic representation of the futile cycling arising from photoadduct decay of the WCC–VVD hetereodimer. Thickness of the arrows indicate the relative rate of reactions. (**B**) Model simulations show responsiveness to increasing light stimuli is maintained via the futile cycle. Upper panel: the rate of photoadduct decay, $l_5$, is decreased by a factor of 10. The first response is identical to the wild type (solid blue) but the system loses responsiveness to increasing light stimuli (dashed blue), while still functioning as an inhibitor of light-mediated WCC activity. Lower panel: the $vvd^-$ mutant loses both the ability to respond to increasing light (similar to the slow-cycling model) and downregulation of the response in constant stimulus. (**C**) Repeated responsiveness is maintained by a sizeable pool of light-activatable WCC (WCC). Simulations show wild-type levels are higher than the $vvd^-$ mutant and the slow-cycling model.

decay of the photoadduct in either protein. Heterodimerization inhibits photoactivated WCC and protects it from degradation, whereas photoadduct decay releases it directly back into the light-activatable pool WCC (Figure 6A).

To demonstrate the key role of this futile cycle, we introduced a variant of the model (the 'slow-cycling model') in which the WCC–VVD complex is artificially stabilized (by slowing down the rate of photoadduct decay, $l_5$ in Figure 2, by a factor of 10). Otherwise, the model is unchanged; in particular, the binding of VVD to WCC is not altered. Indeed, the slow-cycling model fully retains the function of VVD as a competitive inhibitor of WCC. The response to the initial light step is unchanged (Figure 6B) the first spike in the original model (solid blue curve) and the slow-cycling model (dashed blue curve) are identical. However, the responses to subsequent light steps are abolished when photoadduct decay is slowed down; with respect to this feature, the slow-cycling model is very similar to the $vvd^-$ model. Thus, sufficiently rapid photoadduct decay is essential for maintaining the responsiveness of the photosystem.

In order to elicit a significant response, a sizeable pool of light-activatable WCC is required. Our simulations show that this pool diminishes with increasing light (Figure 6C; see also Supplementary Figure S2). However, the wild type always retains a larger pool than the $vvd^-$ mutant, explaining why it remains responsive to subsequent light steps but the mutant does not. Slowing photoadduct decay reduces the light-activatable WCC pool practically to the level of the $vvd^-$ mutant (Figure 6C). This shows that sequestration of WCC by VVD *per se*, which is intact in the slow-cycling model, does not replenish the light-activatable WCC but futile cycling is needed. In particular, note that the light-activatable WCC pool

cannot be fueled by dissociation of the WCC–VVD heterodimer (reaction with rate constants $l_1$ and $l_{-1}$ Figure 2), because this reaction carries a forward net flux during the adaptation phase (newly synthesized VVD captures active WCC) and thereafter equilibrates (net flux $\sim 0$).

To further examine this mechanism, we constructed a simplified model representing only the WCC and VVD components and their key interactions. This model recapitulates the adaptation behavior seen in the full model and the experiments, and, more generally, shows adaptation and maintained responsiveness, provided that VVD is produced at a sufficiently large rate and the WCC–VVD complex decays at a sufficiently large rate to fuel the pool of activatable WCC (Supplementary Text S2). This further demonstrates that indeed downregulation and repeated responsiveness are characteristics that arise from the futile cycle.

Thus, we have identified a futile cycling mechanism in the light-sensing network that is capable of turning feedback inhibition by VVD into sensory adaptation. Binding of VVD to WCC inhibits the WCC, whereas the dissociation of the WCC–VVD complex caused by photoadduct decay continuously replenishes the light-activatable WCC pool and retains the responsiveness of the photosystem in constant light. This regulatory 'motif' allows VVD to act as both an inhibitor and a positive regulator of the light response.

## Futile cycling maintains a light-activatable WCC pool at steady state

The futile cycling model implies that VVD maintains a pool of activatable WCC at steady state in constant light. Therefore, we asked whether we could measure a functional correlate of this

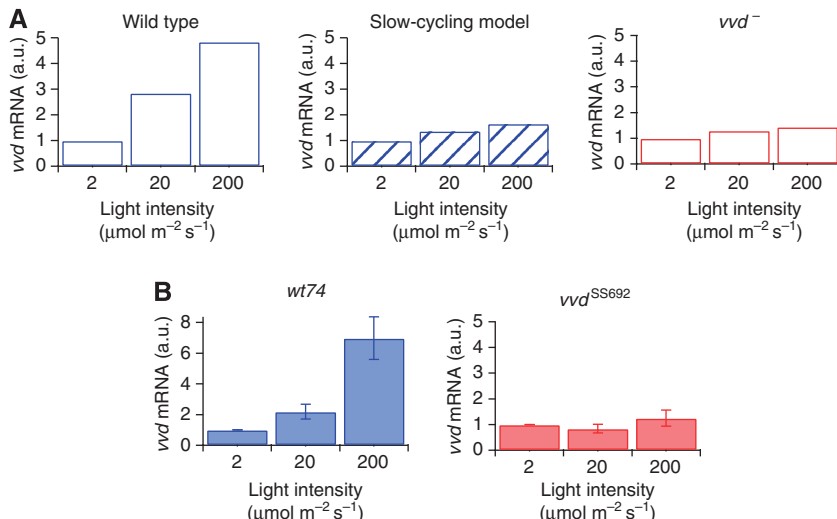

**Figure 7** VVD tracks ambient light levels. (**A**) Model simulations show wild-type *vvd* mRNA levels rise up to fivefold with increasing light intensity. The simulated levels from both the slow-cycling model and the *vvd⁻* mutant remain constant over the three light intensities. All levels are normalized to the respective mRNA values at 2 μmol m⁻² s⁻¹. (**B**) Experimental results confirm the model simulations. *Neurospora* cultures of the indicated strains were grown for 48 h in constant light at 2, 20 and 200 μmol m⁻² s⁻¹, respectively. Samples were collected and mRNA was measured via qRT–PCR. Results shown are from at least two independent experiments measured in triplicates. Source data for this figure is available on the online supplementary information page.

pool. The rate of WCC-dependent gene transcription does not return to zero in constant light but settles to a basal value (*cf.* Figure 1 B–D). This indicates the existence of a light-activatable WCC pool, which allows steady-state gene expression. Focusing on the *vvd* gene as a WCC target, we simulated the steady-state level of *vvd* mRNA in the model, finding that it actually increases with light intensity (Figure 7A; fivefold increase from 2 to 200 μmol m⁻² s⁻¹). This increasing steady-state response in the model is due to a slowly decreasing size of the light-activatable WCC pool (*cf.* Figure 6) and the strong rise in light intensity, as the product of both controls the concentration of the transcriptionally active WCC. Indeed, *vvd* mRNA levels are directly related to the maintenance of the activatable WCC pool by VVD, because in the slow-cycling model the increase of *vvd* transcription with ambient light is much weaker (twofold increase from 2 to 200 μmol m⁻² s⁻¹) and barely noticeable in the absence of VVD.

To test this prediction, *Neurospora* cultures were grown for 48 h at constant light to ensure that a steady state of the photosystem is reached. In the wild type, we observed the expected increase of *vvd* mRNA with light intensity (Figure 7D; see also Malzahn *et al*, 2010). By contrast, no significant change of *vvd* transcription occurred in the *vvd⁻* mutant (Figure 7E), confirming the model prediction.

# Discussion

Sensory adaptation comprises two components: the down-regulation of the response in the presence of a constant stimulus and continued sensitivity of the sensory system to an increase in the stimulus. We have shown here that *Neurospora* light sensing exhibits *bona fide* adaptation and provides an underlying mechanism. Surprisingly, the negative and positive regulatory components of adaptation are mediated by a single molecule, the blue-light photoreceptor VVD. Our model

suggests that the downregulation of the light response could efficiently be achieved by competitive heterodimerization of light-activated VVD with light-activated WCC (feedback inhibition) (Malzahn *et al*, 2010). At the same time, VVD can maintain the responsiveness to further increases in light intensity by channeling active WCC* away from degradation (of the WC-1 component) back into the light-activatable WCC pool (replenishment). This novel mechanism, requiring futile cycling through photoadduct formation and decay, could explain why VVD itself is a blue-light photoreceptor, rather than a light-independent inhibitor of the WCC.

A well-understood example of sensory adaption is vertebrate photoreception, which allows us to perceive objects at nearly constant contrast despite changes over many orders of magnitude in the level of ambient illumination (Fain *et al*, 2001). Most other sensory processes exhibit this type of behavior at least in some range of ambient stimulus intensity. In particular, photoadaptation has been studied for the entrainment of the circadian clock in mammals, with sophisticated light-stimulus protocols (Nelson and Takahashi, 1999; Kronauer *et al*, 1999; Rimmer *et al*, 2000). Much theoretical work on adaptation has focused on the precise downregulation of the response to prestimulus level ('exact' or 'perfect' adaptation; Knox *et al*, 1986; Yi *et al*, 2000; Behar *et al*, 2007; Ma *et al*, 2009); and this concept has informed recent experimental work (Muzzey *et al*, 2009; Houser *et al*, 2012). By comparison, the mechanisms that maintain responsiveness of a sensory system after downregulation of the acute response are much less well understood (e.g., Knox *et al*, 1986; Tang and Othmer, 1994; Behar *et al*, 2007).

Here we have found for *Neurospora* that light adaptation is not perfect. There is a remaining basal response, suggesting that perfect light adaptation is not of overriding physiological importance for this organism. Rather, the photosystem maintains its responsiveness within the natural span of light intensities, so that *Neurospora* perceives relative changes in

intensity (Schwerdtfeger and Linden, 2003; Chen *et al*, 2010; Malzahn *et al*, 2010). This property is likely to be physiologically important for sessile organisms populating habitats with a wide range of ambient light intensities, as it ensures robust responses to the day–night (i.e., sunlight–moonlight) cycles irrespective of the actual habitat and support accurate entrainment of the clock (Malzahn *et al*, 2010; Thommen *et al* 2010; Tsumoto *et al*, 2011).

The data-driven model demonstrates that VVD mediates both downregulation and the maintained responsiveness of the photosystem. In principle, downregulation could be achieved through selective degradation of activated WCC. Indeed, light-activated WC-1 is degraded rapidly (Malzahn *et al*, 2010). However, the level of WC-1 in light is actually higher in the wild type than in adaptation-deficient mutants, arguing against a role for WC-1 degradation in adaptation. Of note, Kronauer *et al* (1999) have previously suggested a simple model of light adaptation where the rate of activation of a single light sensor is postulated as the output driving the (mammalian) circadian clock. Unlike this conceptual model, the model presented here is fully based on experimentally implicated molecular mechanisms, and its functional output is the concentration of the active WCC transcription factor.

Maintained responsiveness of the system requires the replenishment of the WCC pool. Enhanced WC-1 synthesis would be the most direct method. However, this would not be dependent on VVD, contradicting our finding that the *vvd⁻* mutant cannot respond to multiple increases in light intensity. Moreover, WC-1 synthesis is insufficient to compensate its light-induced degradation (Malzahn *et al*, 2010 and Supplementary Figure S2). Instead, we propose that heterodimerization by VVD not only downregulates light responses, but also has a prominent role through the subsequent rechanneling of the WCC back into the light-activatable pool. It is important to note that dissociation of the heterodimer complex cannot fulfill this function, because the net flux is in the direction of the WCC–VVD association to mediate inhibition of the WCC. Rather, the decay of the photoadduct, which resets the individual components to their inactivated forms, is the driving mechanism. The model accounts for the fact that the photocycle of the VVD LOV domain lies within the range of hours (Zoltowski *et al*, 2009). Decay rates for the WCC photoadduct are of the same order of magnitude as the VVD photoadduct (Malzahn *et al*, 2010). The model demonstrates that such a VVD-dependent 'passive' method of replenishment explains the differences in light adaptation between the wild type and the mutant, whereas the 'active' method (*de novo* synthesis) does not.

Important insights into circadian oscillators and adaptation have been gained from simplistic models that allow quite general analysis (e.g., Leloup *et al*, 1999; Leloup Goldbeter, 2000; Akman *et al*, 2008, 2010; Behar *et al*, 2007; Ma *et al*, 2009). Our analysis indicates that certain mechanistic principles might be missed by such conceptual analyses. For example, previous computational searches for network motifs for adaptation did not allow for inhibition by sequestration and, thus, could not come up with the futile-cycling motif described here (Behar *et al*, 2007; Ma *et al*, 2009). Iteration between theory and experiment clearly provides a complementary strategy to establish which mechanistic details matter for biological function (Boothby, 2009). Importantly, the model has been parameterized using the light-step protocol (Figure 1) and subsequently been tested successfully against diverse experimental data (Figures 5 and 7). Thus, we find that the parameters of VVD expression, degradation and heterodimerization with WCC, as well as the rate of replenishment of the active WCC pool by decay of the WCC–VVD complex are constrained by the experimental data, allowing us to identify the dual role of VVD for photoadaptation. Other parameters, in particular the ones pertaining to the action of FRQ, cannot be reliably estimated from our data. This is very likely to be of minor importance in the present context, as data by us and others indicate that VVD has the key role in photoadaptation (Chen *et al*, 2010; Hunt *et al*, 2010; Malzahn *et al*, 2010) However, given FRQs function in the mechanism of the clock our findings imply that further targeted measurements under well-defined conditions will be needed to arrive at quantitatively predictive models of the *Neurospora circadian* clock that link molecular mechanism to physiology (Tseng *et al*, 2012).

Finally, we note that the futile-cycling motif for adaptation could be realized through other mechanisms than photoadduct formation and decay. For example, the regulation of inhibitor affinity by phosphorylation and dephosphorylation (or any other reversible covalent modification) could realize the same regulatory motif, indicating that it might be employed more widely.

## Materials and methods

### *Neurospora* strains and culture conditions

*Neurospora* strains used in this study were *wt74* and the *vvd* loss-of-function mutant *vvd^{SS692}* (Heintzen *et al* 2001). Standard growth medium contained 2% glucose, 0.5% ʟ-arginine, 1 × Vogel's and $10 \, ng \, ml^{-1}$ biotin. Cultures were incubated at $25\,°C$. The light-step protocol (Figure 1A) accommodates one long light step ($20 \, \mu mol \, m^{-2} \, s^{-1}$) to accurately monitor the return to steady state while keeping the overall duration below 24 h to prevent nutrient depletion.

### RNA analysis

Total RNA was extracted with the peqGOLD TriFast reagent (Peqlab). cDNA was synthesized from $1 \, \mu g$ total RNA with the QuantiTect Reverse Transcription Kit (QIAGEN). Transcript levels were quantified by RT − PCR with TaqMan Probes in a StepOne system (Applied Biosystems). Triplicate reactions ($20 \, \mu l$) containing cDNA equivalent to $0.05 \, \mu g$ RNA were analyzed. Primers and probes for measuring *actin*, *frq* and *wc-1* RNA are described elsewhere (Malzahn *et al*, 2010). For the analysis of *vvd* RNA, the following primers and probes were used: VVDII fwd: 5′-TACCCTACCAACACACTGTCTCATAAC-3′; VVDII rev: 5′-TCAACCTCCCATGGGTTCAT-3′; VVDII Probe: 5′-CATGAGCCA TACCGTGAACTCGAGCAC-3′.

### Mathematical modeling and parameter estimation

We used a Markov chain Monte Carlo and Bayesian inference technique, implemented in Fortran 95, to fit the model parameters. Further details of the fitting technique are explained in the Supplementary Information. The model was fitted to the light induction experiments (Figure 1) and the system of ODEs was solved using the Matlab solver ode15s.

## Supplementary information

## Acknowledgements

MB and TH are members of CellNetworks. This work was supported in part by the Initiative and Networking Fund of the Helmholtz Association within the Helmholtz Alliance on Systems Biology/ SBCancer (TH). We thank Sabine Schultz for technical assistance and Andreas Raue for help with the profile likelihood analysis.

*Author contributions:* EG, MB and TH conceived the study. EG performed the modeling. ACRD carried out the experiments. EG, ACRD, MB and TH analyzed the data. EG, MB and TH wrote the paper.

## Conflict of interest

The authors declare that they have no conflict of interest.

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
