## [Review Process File · Molecular Systems Biology]

The *Neurospora* photoreceptor VIVID exerts negative and positive control on light sensing to achieve adaptation

Elan Gin, Axel C. R. Diernfellner, Michael Brunner, Thomas Höfer

Corresponding author: Thomas Höfer, German Cancer Research Center

Review timeline:

Submission date:	26 October 2012
Editorial Decision:	19 December 2012
Revision received:	18 March 2013
Accepted:	18 April 2013

Editor: Thomas Lemberger

Transaction Report:

1st Editorial Decision

19 December 2012

Thank you again for submitting your work to Molecular Systems Biology. We have now heard back from the two referees who agreed to evaluate your manuscript. As you will see from the reports below, the referees find the topic of your study of potential interest. They raise, however, several concerns on your work, which should be convincingly addressed in a revision of this work. The recommendations provided by the reviewers are very clear in this regard.

We would also kindly ask you to include the 'source data' from Figure 1 and 7B (see <http://www.nature.com/msb/authors/index.html#a3.4.3> for guidelines).

If you feel you can satisfactorily deal with these points and those listed by the referees, you may wish to submit a revised version of your manuscript. Please attach a covering letter giving details of the way in which you have handled each of the points raised by the referees. A revised manuscript will be once again subject to review and you probably understand that we can give you no guarantee at this stage that the eventual outcome will be favorable.

REFeree REPORTS:

Reviewer #1 (Remarks to the Author):

Gin et al. study how light affects the circadian clock of *Neurospora*. In particular to the photoactivation of the White Collar Complex (WCC), VIVID (VVD) is also activated by light. VVD can bind to the WCC and inactivate it. The authors use this WCC/VVD interaction to

understand photoadaptation in the *Neurospora* circadian clock, which allows the *Neurospora* circadian clock to function accurately in many lighting conditions. Through modeling and experimental validation, a motif based on a "futile cycle" is tested and presented as a novel mechanism of adaptation.

I think the topic and the combined experimental/modeling approach will be of interest to the readers of MSB. However, many aspects of the manuscript, particularly related to the mathematical modeling, need to be strengthened or clarified. The authors also need to carefully compare their results to previous work on photoadaptation in circadian rhythms. The motif of WCC/VVD could present an interesting, novel motif, but it is not fully explored.

Major Concerns:

- 1) Photoadaptation has been widely studied in circadian rhythms, particularly in mammals. This previous work needs to be carefully considered. Consider discussing some experimental work in higher organisms (e.g. Nelson-Takahashi in rodent, or Rimmer et al. in humans). Previous modeling work should definitely be discussed. The most important study is likely Kronauer et al., *Journal of Biological Rhythms*, 1999, which shows how circadian photoadaptation can be achieved with just one photoreceptive molecule. Could the motif presented in Kronauer et al. have also reproduced the experimental results (further modeling work could test this)? The model also contains a "futile cycle" of activation and inactivation of the photoreceptor, and mathematical analysis shows how activation and reversion to inactivated forms determine the properties of adaptation.
- 2) I am not sure if the data accurately determine the model. For example, previous studies have typically required testing more than one duration of light pulse or step to determine the dynamics of just one photoreceptive molecule (e.g. see Rimmer et al. *American J Physiol.* 2000). Here the protocol seems to be designed somewhat haphazardly. For example, in figure 1, why is the first light level presented for 5 hours and the next for over 10 hours? The shortest light level presented is ~ 5 hours, which seems like a missed opportunity. How did you choose when to take experimental timepoints? How well does the model fit the data (please show the data and model predictions, e.g. Fig 1 and 3, on the same graph)? Can this data actually find the kinetics of both VVD and WCC or are more experiments needed? Can we rule out other mechanisms (e.g. a role of VVD in preparing activated WCC so it can be light sensitive again)?
- 3) If the authors wish to present their simulations as a novel motif, further mathematical analysis is needed. The systems biology community will likely be skeptical about "properties" of a motif that are based on a few simulations with a fixed set of parameters.
- 4) The mathematical model is ill justified. For example:
 - a. Parameters in the equations of the supplement and the model diagram do not match. For example is n_f the same as n_{14} ?
 - b. The variable names are not consistent. The manuscript discusses a dark-form of WCC in the model, but such a variable does not appear in the model equations.
 - c. The model justification is terse and many important details are left out. More explanation is needed. For example, why are Michaelis-Menten or Hill expressions used in some places, but not others? What is your interpretation of a Hill Coefficient of 1.56? What are the differences between the first and third term in the $d[mFRQ]/dt$ equation?
 - d. Please note MSB's policy on providing machine-readable code.
- 5) The circadian part of the model is barely discussed. What is the mechanism of rhythm generation in the model? How does it compare with previous models? How well can you justify the 10-fold change of *frq* expression when rhythms are studied in DD rather than in the presence of light (Figure S3)? More comparison with phase response curves would be helpful.

Minor

1. Page five, remove "(OTHERS?)" from the references cited.
2. Please use correct labels for tables in supplementary texts. That is, please use Table S1 instead of Table1.
3. In discussion, could you provide more explanation on why "adaptations by degradation would be incompatible with longer term memory"?
4. Please remove the statement "The model recapitulates the data." on page 6 and replace it with a

more quantitative statement.

5. It would be helpful to color-code the terms of the equations based on the colors in Figure 2.

Reviewer #2 (Remarks to the Author):

This manuscript presents a moderate amount of new data and a mathematical model to explain the new and previously-published experimental observations on light adaptation in *Neurospora*. This is an original contribution that will be of interest to researchers in several fields, including photobiology, chronobiology, signal transduction, and systems biology. It unifies many puzzling observations about the effects of VVD on light responses. No such detailed model has previously been presented.

The manuscript is well-written in excellent English and does not need detailed editing. The explanations are clear and the model is comprehensible with a complete list of equations and parameter values. The experiments have been conducted at a high standard; the careful calibration and quantitation of the qRT-PCR data is commendable.

Three minor editorial points:

- a. In the second paragraph of Results, a list of references is incomplete (OTHERS).
- b. Figure legend S2: The phrase "10 time lower than" is mathematically impossible. It should be "one tenth of".
- c. Supplementary Text1, paragraph 2. Transcription: "Transcription of *wc-1* in the dark is independent of the WCC, and in the light, by the homodimer." This is confusing. Does it mean independent of the homodimer?

I have three substantial points that should be addressed by the authors.

1. In common with many mathematical modeling papers, the conclusions are stated as certain proofs. For example, the abstract states "Our model demonstrates that the downregulation of the light response is efficiently achieved by ..." No model can demonstrate that a particular molecular mechanism does in fact operate in the organism. All a model can do is suggest a potential mechanism. All of these statements should be qualified to "suggest" that such a mechanism "could" operate or "might" be the right mechanism.
2. Top of page 7: "In summary, the mathematical model reproduces salient features of *Neurospora* photoadaptation." I agree that the model is very successful, and surprising so, for many features. However, in looking at Fig. 1, to my eye one of the most salient features is the drop in all three mRNA levels in the *vvd* mutant immediately after the second and third light steps. This is a surprising and counter-intuitive finding. It is not reproduced by the model, and it suggests there is some other rapid down-regulation of transcription in response to light that is not included in the model. The authors should at least acknowledge that this feature is still unexplained, or provide an explanation.
3. For Fig. S3, the model had to be modified to produced oscillations, and other parameters were changed to preserve light adaptation. This is a troubling statement. The modified model should be the one that was used for all the simulations in the paper. Introducing time delays into the FRQ kinetics could have large effects on the model behavior.

Detailed response to reviewers' comments

All changes in the text are indicated by blue ink.
Summary of changes to the supplementary materials

Previously	Additions
Suppl. Text S1	Revised and Section Maximum likelihood estimation and profile likelihood analysis
Suppl. Fig. S1	Unchanged
Suppl. Fig. S2	Unchanged
Suppl. Fig. S3	Expanded
Suppl. Fig. S4	New Suppl. Text S2 on analysis of simplified futile cycle model. Fig. S6 is updated version of previous Fig. S4; Figs. S7-9 are new.

Comments to reviewer #1

Gin et al. study how light affects the circadian clock of Neurospora. In particular to the photoactivation of the White Collar Complex (WCC), VIVID (VVD) is also activated by light. VVD can bind to the WCC and inactivate it. The authors use this WCC/VVD interaction to understand photoadaptation in the Neurospora circadian clock, which allows the Neurospora circadian clock to function accurately in many lighting conditions. Through modeling and experimental validation, a motif based on a "futile cycle" is tested and presented as a novel mechanism of adaptation.

I think the topic and the combined experimental/modeling approach will be of interest to the readers of MSB. However, many aspects of the manuscript, particularly related to the mathematical modeling, need to be strengthened or clarified. The authors also need to carefully compare their results to previous work on photoadaptation in circadian rhythms. The motif of WCC/VVD could present an interesting, novel motif, but it is not fully explored.

We thank the reviewer for his/her positive evaluation of our manuscript and the constructive criticism. As requested, we have focused our revisions on a more comprehensive characterization of both the full mathematical model of VVD action and the simplified model of the futile-cycling motif for achieving photoadaptation. In the following, this will be detailed in our responses to the specific points raised by the reviewer.

Major Concerns:

1) Photoadaptation has been widely studied in circadian rhythms, particularly in

mammals. This previous work needs to be carefully considered. Consider discussing some experimental work in higher organisms (e.g. Nelson-Takahashi in rodent, or Rimmer et al. in humans). Previous modeling work should definitely be discussed. The most important study is likely Kronauer et al., Journal of Biological Rhythms, 1999, which shows how circadian photoadaptation can be achieved with just one photoreceptive molecule. Could the motif presented in Kronauer et al. have also reproduced the experimental results (further modeling work could test this)? The model also contains a "futile cycle" of activation and inactivation of the photoreceptor, and mathematical analysis shows how activation and reversion to inactivated forms determine the properties of adaptation.

We thank the reviewer for pointing out these relevant papers. We refer now to them in the revised version. In particular, we state in the first paragraph of the introduction (page 3): *"The idea of a light-processing unit has previously been suggested on theoretical grounds for the human circadian clock (Kronauer et al., 1999)."* The other two references are added at the appropriate place in the discussion (page 11): *"In particular, photoadaptation has been studied for the entrainment of the circadian clock in mammals, with sophisticated light-stimulus protocols (Nelson and Takahashi, 1999; Kronauer et al., 1999; Rimmer et al., 2000)."*

Indeed, the model by Kronauer et al. (1999) affords an interesting comparison with our approach, which we added to the Discussion (page 11): *"Of note, Kronauer et al. (1999) have previously suggested a simple model of light adaptation where the rate of activation of a single light sensor is postulated as the output driving the (mammalian) circadian clock. Unlike this conceptual model, the model presented here is fully based on experimentally implicated molecular mechanisms, and its functional output is the concentration of the active WCC transcription factor."*

In our view, interpreting the model by Kronauer et al. as photoadaptation "with just one photoreceptive molecule" relies on postulating that the rate of activation of this molecule is the output driving the photoresponse (which the authors do and which might be the case in the specific system they studied). This is certainly not the case in our system, where the concentration of the WCC transcription factor is the natural output. A first-order ordinary differential equation, such as in Kronauer's model, cannot exhibit adaptation of a concentration variable to a constant input; this follows from the uniqueness of the solution of the initial value problem. The minimal number of independent variables (= molecular species) needed for adaptation of a concentration variable governed by a system of o.d.e. is two.

2) I am not sure if the data accurately determine the model. For example, previous studies have typically required testing more than one duration of light

pulse or step to determine the dynamics of just one photoreceptive molecule (e.g. see Rimmer et al. American J Physiol. 2000). Here the protocol seems to be designed somewhat haphazardly. For example, in figure 1, why is the first light level presented for 5 hours and the next for over 10 hours? The shortest light level presented is ~ 5 hours, which seems like a missed opportunity. How did you choose when to take experimental timepoints? How well does the model fit the data (please show the data and model predictions, e.g. Fig 1 and 3, on the same graph)? Can this data actually find the kinetics of both VVD and WCC or are more experiments needed? Can we rule out other mechanisms (e.g. a role of VVD in preparing activated WCC so it can be light sensitive again)?

This comment refers the accuracy of parameter determination from the data and the experimental light-step protocol, which we will address in turn:

The idea of the experimental protocol is to evoke the full response to a given light intensity and then observe the return of the photosystem to the steady state. This protocol is adequate for the somewhat limited accuracy of the biochemical measurements and has become widely used in the *Neurospora* field. We have explained this now more clearly in the paper on page 5: *The prolonged exposure to constant light intensity was chosen to quantify the transient light response and subsequent return to a steady state.* To our knowledge, we present the most comprehensive experiment of this kind in *Neurospora* to date, comparing the *vvd⁻* mutant systematically with the wildtype (three light steps, quantitative, time-resolved measurements of *vvd*, *wc1* and *frq* mRNA). We note the positive evaluation of the experiments by reviewer #2: *“The experiments have been conducted at a high standard; the careful calibration and quantitation of the qRT-PCR data is commendable.”*

We agree that the timing of the light steps requires further explanation. We chose to leave the cultures longer at the middle light intensity before the shift to high light to show that *vvd*, *frq* and *wc1* mRNA levels in both wildtype and *vvd⁻* mutant stay at light-adapted levels over a longer period of time and do not change significantly. Doing this at all light intensities would not have been possible because liquid *Neurospora* cultures must not be too old before harvesting to avoid unwanted effects through depletion of nutrition. We therefore added the following explanation to the Materials and methods section (page 13): *“The light-step protocol (Fig. 1A) accommodates one long light step ($20 \mu\text{mol m}^{-2}\text{s}^{-1}$) to accurately monitor return to steady state while keeping the overall duration below 24 h to prevent nutrient depletion.”*

Note that to quantify the steady-state levels of *vvd* mRNA accurately at all three light intensities used, we had performed separate experiments with each light level; the data from these experiments are shown in Fig. 7B.

To address the accuracy of parameter estimation, we have originally applied a Bayesian Markov-Chain Monte-Carlo procedure that is adequate for scanning the

parameter space if the model is not fully identifiable. We have now carried out extensive additional work on parameter fitting, using a complementary 'frequentist' profile-likelihood approach (new section in Supplementary Text S1: *Maximum likelihood estimation and profile likelihood analysis*, p.11-16). This approach confirms that key parameters related to the dynamics of VVD and WCC are constrained by the data, while other model parameters cannot be reliably estimated with the available data (new Supplementary Figure S5). In particular, the rate of replenishment of the active WCC pool from the VVD-WCC complex is bounded below (Fig. S5A), providing further support for the futile-cycling model of adaptation. Moreover, the expression and degradation rates of VVD as well as its heterodimerization rate with active WCC are constrained by the data (Fig. S5D,E and S5B,C, respectively). As requested, we have also overlaid a representative maximum-likelihood fit with the experimental data (new Supplementary Figure S4). To discuss these results, we have now added the following new part to the Discussion (page 12): *"Thus we find that the parameters of VVD expression, degradation and heterodimerization with WCC, as well as the rate of replenishment of the active WCC pool by decay of the WCC-VVD complex are constrained by the experimental data, allowing us to identify the dual role of VVD for photoadaptation. Other parameters, in particular the ones pertaining to the action of FRQ, cannot reliably estimated from our data. This is very likely to be of minor importance in the present context, as data by us and others indicate that VVD play the key role in photoadaptation (Chen et al, 2010; Hunt et al, 2010; Malzahn et al, 2010) However, given FRQs function in the mechanism of the clock, our findings imply that further targeted measurements under well-defined conditions will be needed to arrive at quantitatively predictive models of the Neurospora circadian clock that link molecular mechanism to physiology (Tseng et al., 2012)."*

3) If the authors wish to present their simulations as a novel motif, further mathematical analysis is needed. The systems biology community will likely be skeptical about "properties" of a motif that are based on a few simulations with a fixed set of parameters.

We are grateful to the reviewer for asking for further mathematical analyses on the reduced model. Our additional work (detailed in Supplementary Text S2) has now provided more rigorous support for the conclusion that futile cycling can maintain the responsiveness of the *Neurospora* photosystem. First, we have directly fitted the reduced model of the futile-cycling motif directly to the experimental data of Fig. 1 (instead of choosing a parameter set "by hand"), achieving semi-quantitative agreement (new Supplementary Fig. S6). We then asked how this fit constrains the model parameters, using the profile-likelihood method (new Supplementary Fig. S7). Most importantly, this analysis shows that the rate of replenishment of the active WCC pool by WCC-VCC photoadduct decay (futile cycling) is well constrained by the data (Fig. S7A). Specifically, this

rate must not be too small to accommodate the data. This conclusion is further strengthened by a second approach, constrained optimization. Here we did not use the experimental data but asked, more generally, which parameter values the reduced model would adopt if we required it to show repetitive spikes to a 'staircase' input. We minimized the deviation of the steady state response from the baseline (i.e., adaptation) and required, as a constraint, that each step increase in the light input should trigger a new spike nearly as large as the preceding one. Again using the profile-likelihood method, we found that this constraint could only be met if the futile cycling rate was above a certain threshold (Fig. S9A). Other parameters are also constrained, showing that VVD synthesis rate must exceed a certain threshold (Fig S9D), and the degradation rate of the VVD-WCC complex must not be too large (Fig. S9F). Degradation of active WCC must be sufficiently large to control its concentration (Fig. S9G). Thus both the fitting of the futile-cycling motif to the experimental data and the more general analysis of the conditions under which this motif produces adaptation and maintained responsiveness support the key role of the futile cycling rate.

Accordingly, we modified the corresponding section in the main text: *"To further examine this mechanism, we constructed a simplified model representing only the WCC and VVD components and their key interactions. This model recapitulates the adaptation behavior seen in the full model and the experiments and, more generally, shows adaptation and maintained responsiveness provided that VVD is produced at a sufficiently large rate and the WCC-VVD complex decays at a sufficiently large rate to fuel the pool of activatable WCC (Supplementary Text S2). This further demonstrates that indeed downregulation and repeated responsiveness are characteristics that arise from the futile cycle."*

4) The mathematical model is ill justified. For example:

a. Parameters in the equations of the supplement and the model diagram do not match. For example is n_f the same as n_{14} ? This is indeed the case and this has been corrected in equations.

b. The variable names are not consistent. The manuscript discusses a dark-form of WCC in the model, but such a variable does not appear in the model equations. We refer to the inactive form of the WCC also as the 'dark form'. We have included this description in the table of variables, Table S2.

c. The model justification is terse and many important details are left out. More explanation is needed. For example, why are Michaelis-Menten or Hill expressions used in some places, but not others?

Saturation terms have naturally been used for enzymatic reactions: induction of gene expression and phosphorylation steps by FRQ. Photoactivation, complex formation, dissociation and (non-enzymatic) photoadduct have been modeled by

mass action terms. This is now explained in Supplementary Text S1.

What is your interpretation of a Hill Coefficient of 1.56?

Hill coefficients are to be viewed as effective quantities that are not directly related to molecular mechanism. Hence fractional Hill coefficients are standard (this had already been noted by Archibald Hill when suggesting the Hill equation for oxygen binding by hemoglobin).

What are the differences between the first and third term in the $d[mFRQ]/dt$ equation?

We have included explanation of this in the Mathematical model section of Supplementary Text1:

In the dark, the inactive, dark form of the WCC can drive expression of frq by binding to the clock-box (C-box) element of the frq promoter (Froehlich et al., 2003). This is described in the second term in the equation for the $d[mFRQ]/dt$, where WCC binds with affinity K_{14} . Additionally, light-activated WCC (in the form of the homodimer, WCC^-WCC^*) can also bind to the C-box and to the proximal light-responsive element (LRE) of frq. Binding affinity of WCC^*-WCC^* to the C-box is given by $K_{14'}$, while WCC^*-WCC^* binds to the LRE with affinity given by K_{13} .*

d. Please note MSB's policy on providing machine-readable code. We will supply Matlab code for all simulations in the Supplementary.

5) The circadian part of the model is barely discussed. What is the mechanism of rhythm generation in the model?

First, we would like stress that the purpose of this paper has been to develop a detailed model of the photoadaptation module in *Neurospora* rather than a model of the *Neurospora* circadian clock. We extended the photoadaptation model by a simplified description of the negative feedback loop that is thought to be responsible for generating circadian oscillations in the dark to strengthen our argument that such a separation of the core clock mechanism and the photoadaptation module is possible. Indeed, in agreement with experimental data, the lack of VVD does not affect the free-running oscillations but only modulates their onset. This is clearly shown in Supplemental Fig. S3.

The mechanism of rhythm generation is assumed to be delayed negative feedback of the WCC target FRQ on the activity of the WCC. We have stated this now more clearly in the legend of Fig. S3: “*Self-sustained oscillations in constant darkness are thought to arise through delayed negative feedback of the WCC target FRQ on the activity of the WCC (Brunner and Kaldi, 2008; Baker et al, 2012).*”

How does it compare with previous models?

The most detailed mathematical model to date is also based on this mechanism (Tseng et al., 2012).

How well can you justify the 10-fold change of frq expression when rhythms are studied in DD rather than in the presence of light (Figure S3)?

This is consistent with experimental data. The amplitude of FRQ induction in light is ~10fold larger than in constant darkness (Crosthwaite et al. Cell 81, 1003-1012 and our own unpublished data).

More comparison with phase response curves would be helpful.

We calculated the PRC of the extended model and found the shape consistent with published experimental data (see Crosthwaite et al. Cell 81, 1003-1012, Fig. 4). For ease of reference, we have copied the experimental data (in the redrawn version of Tseng et al., 2012; Fig. 7A) below on the left and the PRC calculated with our model (for smaller and larger light pulses, circles and crosses, respectively) below on the right.

Minor

1. Page five, remove "(OTHERS?)" from the references cited.

Has been done.

2. Please use correct labels for tables in supplementary texts. That is, please use Table S1 instead of Table 1.

Has been corrected.

3. In discussion, could you provide more explanation on why "adaptations by degradation would be incompatible with longer term memory"?

The reason is that the degradation rate would then also set the time scale of WC-1 recovery; degradation is fast, so would be recovery. With VVD, this is not the

case. As this is a minor point compared to the experimental finding arguing against an important role for WC-1 degradation (lower WC-1 in an adaptation-deficient mutant than in the WT), we chose not to elaborate but to delete this statement in the text. Thus we avoid confusion and save some space for the additions to text made during the revision.

4. Please remove the statement "The model recapitulates the data." on page 6 and replace it with a more quantitative statement.

We have rewritten the section on model fitting and comparison with the data in the main text (pages 6 and 7). *"To fit the model to the light-step experiments in Figure 1, we first determined the kinetic parameters for the model without VVD protein, mimicking the vvd⁻ mutant (using both a Bayesian Markov Chain Monte Carlo algorithm and a maximum likelihood estimate; Supplementary Text S1). The available data constrain parameters of the model that govern VVD function (including VVD production, heterodimerization with WCC and photoadduct decay; Supplementary Fig. S5). For a reduced, biologically less detailed version of the model all parameters can be identified from the data (Supplementary Text S2). The parameterized model reproduces the very slow decline of the response after the first light step, caused by degradation of light-activated WCC and inhibition of WCC by FRQ-mediated phosphorylation. The following two light steps do not elicit substantial responses (Fig. 3E-G; cf. Fig. 1E-G). Interestingly, the vvd⁻ mutant data show an immediate drop of mRNA levels after the light pulse that cannot be explained by the present model and hints at additional mechanisms regulating mRNA synthesis or degradation."*

5. It would be helpful to color-code the terms of the equations based on the colors in Figure 2.

This has been done.

Comments to reviewer #2

Reviewer #2 (Remarks to the Author):

This manuscript presents a moderate amount of new data and a mathematical model to explain the new and previously-published experimental observations on light adaptation in Neurospora. This is an original contribution that will be of interest to researchers in several fields, including photobiology, chronobiology, signal transduction, and systems biology. It unifies many puzzling observations about the effects of VVD on light responses. No such detailed model has previously been presented.

The manuscript is well-written in excellent English and does not need detailed editing. The explanations are clear and the model is comprehensible with a complete list of equations and parameter values. The experiments have been conducted at a high standard; the careful calibration and quantitation of the qRT-PCR data is commendable.

We thank the reviewer for the positive evaluation of our findings.

Three minor editorial points:

a. In the second paragraph of Results, a list of references is incomplete (OTHERS).

Corrected.

b. Figure legend S2: The phrase "10 time lower than" is mathematically impossible. It should be "one tenth of".

Has been corrected.

c. Supplementary Text1, paragraph 2. Transcription: "Transcription of wc-1 in the dark is independent of the WCC, and in the light, by the homodimer." This is confusing. Does it mean independent of the homodimer?

We have rewritten this to avoid confusion:

Transcription of wc-1 in the dark is independent of the WCC, given by the basal rate k_{12} . In light, transcription of the WCC is driven by the homodimer.

I have three substantial points that should be addressed by the authors.

1. In common with many mathematical modeling papers, the conclusions are stated as certain proofs. For example, the abstract states "Our model demonstrates that the downregulation of the light response is efficiently achieved by ..." No model can demonstrate that a particular molecular mechanism does in fact operate in the organism. All a model can do is suggest a potential

mechanism. All of these statements should be qualified to "suggest" that such a mechanism "could" operate or "might" be the right mechanism.

We fully agree with the reviewer that results from the model without experimental support should be indicated as such and apologize if this has not been the case in all places. The cited statement is in the Discussion (page 10) and has been corrected: *"Our model suggests that the downregulation of the light response could efficiently be achieved by competitive heterodimerization of light-activated VVD with light-activated WCC (feedback inhibition) (Malzahn et al., 2010)."* We have also rephrased a related claim in the Introduction (page 4) to read: *"We identified a novel 'adaptation motif' in the WCC-FRQ-VVD reaction network and suggest that this motif allows Neurospora to sense relative changes in light intensity."*

2. Top of page 7: "In summary, the mathematical model reproduces salient features of Neurospora photoadaptation." I agree that the model is very successful, and surprising so, for many features. However, in looking at Fig. 1, to my eye one of the most salient features is the drop in all three mRNA levels in the vvd mutant immediately after the second and third light steps. This is a surprising and counter-intuitive finding. It is not reproduced by the model, and it suggests there is some other rapid down-regulation of transcription in response to light that is not included in the model. The authors should at least acknowledge that this feature is still unexplained, or provide an explanation.

The reviewer is right with the observation that there is a drop in RNA levels following the second and, in particular, the third light pulse in the *vvd* mutant strain. We noticed this but as yet have not come up with an explanation. Preliminary analyses suggest that there might be a refractory time for promoters after transcription initiation, which we uncover when synchronizing transcription by a light pulse. We are presently investigating this phenomenon but do not mention this in the text because we could be completely wrong with our hypothesis. We added on page 7: *"Interestingly, the vvd- mutant data show an immediate drop of mRNA levels after the light pulse that cannot be explained by the present model and hints at additional mechanisms regulating mRNA synthesis or degradation."*

3. For Fig. S3, the model had to be modified to produced oscillations, and other parameters were changed to preserve light adaptation. This is a troubling statement. The modified model should be the one that was used for all the simulations in the paper. Introducing time delays into the FRQ kinetics could have large effects on the model behavior.

We agree with the reviewer that the extended model producing the oscillations must respect all the experimental data in the paper on adaptation. This is indeed the case, as we show in an extended version of Supplementary Figure S3. In particular, the time-delayed FRQ kinetics do not 'destroy' the adaptation behavior

associated with the dynamics of VVD.

The purpose of adding self-sustained oscillations in the dark with a simplified model was to show that VVD affects the onset of the oscillations but not the oscillations themselves. The question of a unique parameter set for all variants of the model is a long-term goal of our work and cannot be addressed with the data that is currently available to us. In particular, we do not yet have a data set with WC-1 and FRQ oscillations that is comparable in time resolution to the data we used to parameterize the adaptation model. Therefore, we lack an appropriate experimental basis for parameterizing the larger model (using published data is not an alternative here, as experimental conditions are usually not fully comparable). Moreover, the simple model extension neglects many details of the molecular regulation of FRQ activity and its interaction with WCC (including the nucleo-cytoplasmic shuttling of both proteins). These details may well become important in the context of a larger model, but such a model will need to be the subject of a separate study.

Thank you again for sending us your revised manuscript. We are now satisfied with the modifications made and I am pleased to inform you that your paper has been accepted for publication.

Thank you very much for submitting your work to Molecular Systems Biology.

Reviewer #1 (Remarks to the Author):

The authors have adequately addressed my concerns.

Reviewer #2 (Remarks to the Author):

The authors have adequately addressed the concerns of both reviewers, and the paper is suitable for publication without further revision.